# Multiplex PCR Approach for Rapid African Swine Fever Virus Genotyping

**DOI:** 10.3390/v16091460

**Published:** 2024-09-13

**Authors:** Matthias Licheri, Manon Flore Licheri, Kemal Mehinagic, Emilia Radulovic, Nicolas Ruggli, Ronald Dijkman

**Affiliations:** 1Institute for Infectious Diseases, University of Bern, 3001 Bern, Switzerland; 2Graduate School for Cellular and Biomedical Sciences, University of Bern, 3012 Bern, Switzerland; 3Multidisciplinary Center for Infectious Diseases, University of Bern, 3012 Bern, Switzerland; 4Institute of Virology and Immunology IVI, 3147 Mittelhäusern, Switzerland; 5Department of Infectious Diseases and Pathobiology, Vetsuisse Faculty, University of Bern, 3012 Bern, Switzerland; 6European Virus Bioinformatics Center, 07743 Jena, Germany

**Keywords:** African swine fever virus, genotyping, multiplex PCR, p72, p54, CVR, ECO1, nanopore sequencing

## Abstract

African swine fever virus (ASFV) has been spreading through Europe, Asia, and the Caribbean after its introduction in Georgia in 2007 and, due to its particularly high mortality rate, poses a continuous threat to the pig industry. The golden standard to trace back the ASFV is whole genome sequencing, but it is a cost and time-intensive methodology. A more efficient way of tracing the virus is to amplify only specific genomic regions relevant for genotyping. This is mainly accomplished by amplifying single amplicons by PCR followed by Sanger sequencing. To reduce costs and processivity time, we evaluated a multiplex PCR based on the four primer sets routinely used for ASFV genotyping (*B646L*, *E183L*, *B602L*, and intergenic *I73R-I329L*), which was followed by Nanopore ligation-based amplicon sequencing. We show that with this protocol, we can genotype ASFV DNA originating from different biological matrices and correctly classify multiple genotypes and strains using a single PCR reaction. Further optimization of this method can be accomplished by adding or swapping the primer sets used for amplification based on the needs of a specific country or region, making it a versatile tool that can speed up the processing time and lower the costs of genotyping during ASFV outbreaks.

## 1. Introduction

African swine fever virus (ASFV) is the etiological agent of African swine fever (ASF), which is a highly contagious hemorrhagic disease of domestic pigs and wild boars with a mortality of up to 100% [1]. ASFV was first described in Africa in 1921 where currently 24 different p72 genotypes are described; however, only two genotypes have spread to other continents (i.e., ASFV genotype I and II) [1,2]. ASFV genotype I was introduced in Europe in the late 1950s and then spread to the Caribbean and Brazil [3]. Genotype I continued circulating up to the 1990s, when it was eradicated from all countries outside of Africa with the exception of Sardinia (Italy) [3]. This genotype has newly been reported in pigs with chronic disease in China in 2021 [4]. A highly virulent genotype II ASFV was introduced in Georgia in 2007, subsequently spreading to the neighboring countries, infecting both wild boars and domestic pigs, and reaching the European Union in 2014 [5,6,7]. As of June 2024, ASFV has been infecting animals in 28 European countries, with Sweden, Montenegro, and Albania being the latest countries reporting an outbreak [8]. Eastwards, ASFV was detected in China in June 2018 and spread further to multiple countries in Asia and Oceania [9,10]. Another transcontinental jump resulted in an ASFV outbreak in the Dominican Republic and Haiti in July 2021 [11]. To date, only Vietnam has a commercially available vaccine that protects from homologous infection; however, the global lack of efficient vaccines or treatments makes early detection a key tool to prevent an outbreak from spreading further [1,12].

Sequencing is a powerful tool that enables the phylogenetic analysis of a virus; however, whole genome sequencing (WGS) of ASFV is resource-intensive due to its genome size. Therefore, the amplification of specific genomic regions for genotypic and intra-genotypic classification of ASFV is a commonly used alternative [13]. The 24 different genotypes of ASFV are classified by sequencing the partial sequence of the *B646L* gene coding for the C-terminal region of the p72 viral capsid protein [14,15,16]. For further discrimination between viral strains, other genomic regions are being used, such as the entire *E183L* gene encoding the viral envelope protein, p54, and the central variable region (CVR) within the *B602L* gene with its characteristic tandem repeat sequences [17,18,19]. In contrast, the intergenic region between *I73R* and *I329L* genes is used to cluster ASFV strains geographically and can be used to discriminate genotype II strains [20,21,22]. Using these genomic regions in parallel for genomic surveillance increases the differentiation abilities and allows studying the transmission dynamics in a region or country. Therefore, this approach is being used in multiple countries, such as Poland, Estonia, and China, where multiple molecular markers such as *O174L*, MGF505-*5R*, *K145R*, ECO2, and *EP402R* are combined to improve the traceability of ASFV outbreaks [20,21,23,24]. However, these amplicons are amplified and analyzed individually by gel electrophoresis or Sanger sequencing, both of which can be labor and cost-intensive, especially when analyzing many samples.

Therefore, we generated a low-cost and rapid protocol using the PCR primers recommended by the European reference laboratory for ASFV genotyping, as these allow to broadly identify ASFV genotypes and subtypes and are not too specific to one geographic location or genotype. We used the four amplicons (p72 (*B646L* gene), PPA (*E183L* gene), CVR (*B602L* gene), and ECO1 (intergenic *I73R-I329L*)) in a multiplex format for subsequent amplicon sequencing on the Oxford Nanopore Technologies (ONT) sequencing platform. This allows ASFV strain identification of multiple samples within 5 h [25].

## 2. Materials and Methods

### 2.1. Viral DNA

Viral DNA used for this study was extracted at the BSL-3Ag facility of the Institute of Virology and Immunology IVI in Mittelhäusern, Switzerland, from EDTA blood, serum, or organ homogenate (liver and lymph nodes) from specific pathogen-free pigs previously infected with ASFV Estonia 2014 (pigs #68, 75) or Georgia 2007 (pigs #85, 86, 87, 88, 89, 90). These samples were from in vivo studies performed at the IVI and described elsewhere [26]. These experiments were performed in compliance with the Swiss animal protection law (TSchG SR 455; TSchV SR 455.1; TVV SR 455.163) and were reviewed by the committee on animal experiments of the canton of Bern and approved by the cantonal veterinary authority under the licenses BE18/2019 and BE46/2022. Viral DNA was also extracted from supernatant of porcine macrophages infected with the strains 1819V, Perpignan, KWH or Malawi or from the supernatant of immortalized porcine kidney macrophage cells (IPKM) infected with ASFV Georgia 2007. Strain 1819V is an ASFV of unknown origin that was a cell culture adapted in the diagnostic laboratory of the IVI. The ASFV strains Perpignan, KWH/12 (KWH), and Malawi Lil-20/1 (Malawi) were provided to the IVI by Dr. P. Wilkinson, IAH, Pirbright, UK [27]. ASFV DNA was purified from EDTA blood with the NucleoSpin Blood kit, from serum and cell culture supernatant with the NucleoSpin Virus kit, and organ homogenate with the NucleoSpin DNA RapidLyze kit (Macherey-Nagel, Düren, Germany), according to the manufacturer’s instructions. Following nucleic acid extraction, viral DNA was further processed at BSL2 in accordance with the Swiss containment ordinance (ECOGEN A210161-01 and A192517-01).

### 2.2. qPCR

To determine the viral load in the different samples, we performed a qPCR targeting the *B646L* gene using previously published primers, where we exchanged the fluorescent dye from Cy5 to FAM [28]. Here, 2 μL of ASFV sample was used as input for a 10 μL reaction mixture using the Luna Universal Probe qPCR Master Mix (New England BioLabs (NEB), Ipswich, MA, USA) according to the manufacturer’s instructions. The analysis was conducted on a QuantStudio 7 Flex Real-Time PCR System (Thermo Fisher Scientific, Waltham, MA, USA) with the following cycle profile: initial denaturation step (1 min, 95 °C), succeeded by 45 cycles of denaturation (15 s, 95 °C), annealing and elongation (30 s, 60 °C). The fluorescence intensity was measured after each cycle.

### 2.3. Multiplex PCR

For the generation of a low-cost and rapid identification protocol for ASFV, we used the four primer sets recommended by the European reference laboratory that target the *B646L* (partial amplification of the p72 gene currently used for genotyping), *B602L* (CVR), and *E183L* (PPA) genes, as well as the intergenic region *I73R-I329L* (ECO1) [25]. The primer sets were pooled using different concentrations for each forward and reverse primer ranging from 0.1 to 0.5 µM. The reaction mixture was composed of 5 µL of 5x Q5 Reaction Buffer (New England BioLabs (NEB), Ipswich, MA, USA), 0.25 µL of Q5 Hot Start High-Fidelity DNA Polymerase (NEB), 0.5 µL of 10 mM dNTP mix (Promega, Madison, WI, USA), 6 µL of the primer pool, 2 µL of extracted DNA and topped up to 25 µL with nuclease-free water (Thermo Fisher Scientific, Waltham, MA, USA). The PCR profile started with an initial denaturation step of 30 s at 98 °C followed by 35 cycles of denaturation (10 s, 98 °C), annealing (10 s, 59 °C), elongation (30 s, 72 °C), and a final extension of 2 min at 72 °C. For the Platinum SuperFi II PCR Master Mix (Invitrogen, Waltham, MA, USA), 2 μL of extracted DNA was used as input for a 25 μL reaction composed of 12.5 μL of 2x Platinum SuperFi II PCR Master Mix, 6 μL of primer pool, and 4.5 μL of nuclease-free water. For amplificon, a cycling profile with an initial denaturation step (30 s, 98 °C), 35 cycles of denaturation (10 s, 98 °C), annealing (10 s, 60 °C), and extension (30 s, 72 °C), and a final extension (30 s, 5 min) was used.

### 2.4. Size-Selection and Library Preparation

After PCR amplification, the amplicons were one-way size-selected using the SPRIselect magnetic beads (Beckman Coulter, Brea, CA, USA) in a 0.8x ratio following the SPRIselect Left Workflow described by the producer. Elution was performed in 15 µL of Elution Buffer (Omega Bio-Tek Inc., Norcross, GA, USA). After quantification using the Qubit 1x dsDNA HS Assay Kit (Thermo Fisher Scientific) with a Qubit 4 fluorometer (Thermo Fisher Scientific), 200 fmol of single amplicons or a single pool of multiple amplicons were assigned to an individual sequencing barcode and used as input for nanopore sequencing library preparation using the ligation sequencing kit (SQK-LSK109 combined with EXP-NBD196, or SQK-NBD114.96, ONT, Oxford, UK) and sequenced on a MinION device (Mk1B, ONT) using a MinION flow cell (R9.4.1 or R10.4.1, FLO-MIN106D, ONT) according to the manufacturer’s instructions with real-time high-accuracy basecalling (Guppy v5.1.13 or Dorado v7.2.13, GPU enabled, ONT) enabled in the MinKNOW software (v21.11.8 or 23.11.7, ONT) (Figure 1). Sequencing runs for the initial primer calibration assay, the sensitivity assay, and the sequencing of the clinical samples were all performed independently.

### 2.5. Bioinformatic Analysis

To analyze the final experiment, where the methodology was applied to samples of different genotypes, demultiplexed raw sequence data were used as input for a custom-made bash script piping minimap2 (v2.21-r1071) for the reference-based assembly and medaka (v1.5.0, ONT) for polishing the consensus draft, which was then used as input for further analysis with the Geneious Prime software package (v2022.0.2, Biomatters, Auckland, New Zealand) [29,30]. To call the consensus sequence, we set a minimum nucleotide depth of 10x and 20x when using the SQK-NBD114.96 or the SQK-LSK109 kit, respectively. All the consensus sequences from the individual target regions were aligned using the MAFFT aligner (algorithm: auto; scoring matrix: 200PAM/k = 2; gap open penalty: 1.53; offset value: 0.123). Alignments from the p72 and PPA target region were used to generate phylogenetic trees with the Geneious Tree Builder (genetic distance model: Tamura-Nei; tree-building method: neighbor-joining; 10,000 bootstraps; no outgroup) and further processed using the R package ggtree (v3.8.0) [31]. The alignments generated for the ECO1 and CVR target regions were processed using the R package ggmsa (v1.6.0) [32]. Finally, the heat maps generated to compare the sensitivity were generated using the R package ComplexHeatmap (v2.16.0) [33,34].

## 3. Results

### 3.1. Multiplex PCR Calibration

To optimize the ASFV multiplex PCR assay, we first evaluated the individual primer concentrations ranging from 0.5 to 1.0 µM using viral DNA from a genotype II Georgia 2007 strain that was extracted from an EDTA blood sample of an experimentally infected pig using the Q5 DNA polymerase. After amplicon sequencing, we analyzed the data with the InterARTIC pipeline, using a custom-generated concatenated amplicon primer scheme as a reference file. This allowed us to compare the number of generated reads for each amplicon using a normalized read depth value of 400 [35]. Based on these results, we selected the primer mixture with the following concentrations for further evaluation: 0.35 µM CVR1/2, 0.1 µM p72-U/D, 0.5 µM PPA89/722, and 0.25 µM ECO1A/B.

### 3.2. Evaluation of Detection Sensitivity

To evaluate the detection sensitivity of the multiplex assay against the individual amplicon assay (singleplex) using the Q5 DNA polymerase, we used six 10-fold serial dilutions of an IPKM cell cultured ASFV strain (Georgia 2007) (Figure 2). Given that Platinum SuperFi II DNA polymerase supports the use of multiple primers at a uniform annealing temperature, we also evaluated the multiplex format with this new recombinant DNA polymerase. The mean sequencing depth from three independent replicates revealed a slight increase in sensitivity of approximately 10-fold in the multiplex format compared to the singleplex when using Q5 DNA polymerase for the ECO1, p72, and PPA amplicons. Although the PPA amplicon was detected at the fifth 10-fold serial dilution (Ct value 28.7), the standard deviation indicates a potential drop in sequencing depth below the consensus calling threshold due to one replicate being negative (Appendix A). Detection of the CVR amplicon resulted in a lower sensitivity in the multiplex format with an approximately 7-fold decrease. Interestingly, the multiplex assay using the Platinum SuperFi II DNA polymerase demonstrated an overall higher detection sensitivity than both assay formats with the Q5 DNA polymerase except for the CVR amplicon, which was not amplified. The other amplicons (ECO1, p72, and PPA) had a detection sensitivity increase of approximately 10-fold. This suggests that the multiplex format with the Platinum SuperFi II provides higher overall detection sensitivity compared to Q5 DNA polymerase.

### 3.3. Assessing Multiplex PCR Compatibility with Diverse ASFV Samples

After establishing the detection sensitivity, we evaluated the compatibility of the multiplex assay with different ASFV genotypes using both polymerases and sequenced the generated amplicons. We used DNA extracted from cell culture supernatants of three different strains: Perpignan (genotype I), Malawi (genotype VIII), and KWH (genotype X), as well as one cell-culture adapted ASFV isolate of unknown genotype, which was named 1819V. Our results showed that all four target regions were efficiently amplified using the Q5 DNA polymerase. In contrast, the Platinum SuperFi II DNA polymerase successfully amplified three amplicons (p72, ECO1, and PPA) across all samples with the CVR amplicon amplified only for KWH and 1819V (Table 1). These findings demonstrate that the current multiplex assay format is compatible with different ASFV genotypes.

To evaluate whether this method can be used on nucleic acids isolated from different biological matrices, we extracted ASFV DNA from multiple paired EDTA blood and serum samples, which were collected at the peak of viremia from six ASFV Georgia 2007-infected pigs (No. 85, 86 (only serum), 87, 88, 89, 90). Additionally, we obtained extracted DNA from liver (li) and lymph nodes (LN) biopsies taken during necropsy from pigs (No. 68 and 75, day 17 post-infection) that had been previously experimentally infected with the ASFV genotype II strain Estonia 2014 [26]. We detected all four amplicons in the EDTA and serum samples except for serum sample 85 amplified with the Platinum SuperFi II DNA polymerase, where the CVR amplicon was undetected (Table 1). All four amplicons were successfully sequenced in both liver samples except for the PPA amplicon when using the Q5 DNA polymerase (Table 1). Sequencing of the lymph node samples revealed that the PPA amplicon was missing for 75LN, and no amplicon was detected for 68LN with the Q5 DNA polymerase. In contrast, when using the Platinum SuperFi II DNA polymerase, only the CVR amplicon was undetected for both 68LN and 75LN, demonstrating the higher sensitivity of the assay with this DNA polymerase.

### 3.4. Phylogenetic Analysis of the Target Regions

After confirming that the multiplex assay can amplify the four target regions for different ASFV genotypes, we assessed whether the resulting sequencing information can be used as input for phylogenetic analysis. We combined the sequence data obtained from the *B646L* target region, amplified using both DNA polymerases, with representative GenBank sequences of each of the 24 genotypes to construct a phylogenetic tree. This showed that the consensus sequence from the different sample matrices clustered with their expected genotype except for the lab-adapted strain 1819V. Additionally, sequences generated using both DNA polymerases clustered together (Figure 3). Serum85-SF and Est68LN-SF both had low coverage, resulting in two and one mismatches, respectively, causing them to cluster slightly outside of the ASFV genotype II samples. Further genotyping stratified sequences into subgroups based on the *E183L* sequence using the PPA amplicon data, where sequences from both polymerases cluster together (Figure 4). However, the coverage was too low to generate a consensus sequence for all liver and lymph node samples as well as the serum85-SF sample. In addition to phylogenetic analysis, the *I73R-I329L* intergenic region sequence data can be used as a molecular marker for geographically clustering genotype II ASFV strains based on the presence of a 10 nt tandem-repeat sequence, aligned with the four known variants within ASFV genotype II (Figure 5). Here, the Estonia 2014 sequences amplified with the Platinum SuperFi II DNA polymerase (Est68li-SF, Est75li-SF, Est68LN-SF, and Est75LN-SF) lacked nucleotides in the tandem repeat region compared to those amplified with the Q5 DNA polymerase, indicating the need for a higher coverage to improve the consensus sequence quality. Finally, multiple sequence alignment of the *B602L* amplicon allows for further characterization of the tandem-repeat sequences at both the nucleotide and translated amino acid sequence levels, aiding in resolving phylogenies on a regional level (Figure 6A,B).

In conclusion, this study demonstrates that the current ASFV multiplex PCR assay and sequencing methodology described herein facilitates the correct genotyping and subgrouping of ASFV from different biological matrices. Furthermore, the multiplex approach allowed us to phylogenetically characterize the 1819V despite the p72 amplicon failing to cluster with a specific genotype. Notably, using a different enzyme can enhance the sensitivity of the ASFV multiplex PCR assay.

## 4. Discussion

To our knowledge, this is the first study to demonstrate that the four primer sets recommended by the European reference laboratory for ASFV can be multiplexed in a single PCR reaction and can be subsequently used to discriminate between ASFV genotypes using nanopore sequencing technology.

Currently, sequencing specific ASFV genomic regions is mostly performed using Sanger sequencing, which has a high accuracy (>99%) and is ideal for shorter sequences of up to 800–1000 bp, but it lacks scalability [36]. In our study, we combined the multiplex PCR with ONT sequencing, which is a portable third-generation sequencing technology based on real-time single-molecule long-read sequencing. This enabled the sequencing of four amplicons within one barcode and the multiplexing of multiple samples in a single sequencing run, resulting in shorter turnaround times—approximately 5 h for library preparation and sequencing on the ONT platform compared to one working day for Sanger sequencing. Additionally, the continuous update of sequencing chemistries and basecalling algorithms by ONT has resulted in current base accuracies of Q20 (or 99%) for simplex reads or Q30 (or 99.9%) for duplex reads, reaching accuracies similar to the ones of next-generation sequencing [37,38]. Although the costs per sample are similar for ONT and Sanger, the ability to sequence more genomic regions within one barcode and the possibility of sequencing reads longer than 1000 bp reduce the price per base for ONT. Moreover, the ability to sequence larger genomic regions allows the entire p72 coding region (approximately 2 kb long) to be sequenced, which has been recently used to propose a new classification of ASFV in six genotypes [13]. The WGS of ASFV is the ideal approach for phylogenetic analysis, but it is cost-intensive due to its genome size [13,39]. Therefore, primers amplifying specific genomic regions are often used for ASFV phylogenetic analysis [13,14,15,16,17,18,19,20,21,22,23,24,39]. However, these are often amplified in a singleplex reaction and sequenced individually. In this study, we demonstrate that the primer sets recommended for genotyping by the European reference laboratory for ASFV can be combined in the multiplex format, and that the resulting sequences can be used for the phylogenetic analysis of ASFV. Additionally, our multiplex PCR approach can be complemented or adapted with individually optimized primer pairs relevant for isolates from a specific geographical region (i.e., *O174L*) or with primers amplifying the entire *B646L* gene [13]. This approach, therefore, provides a versatile and high-throughput tool that can be further adapted to the needs of a routine diagnostic laboratory setting for conducting ASFV surveillance.

We demonstrate that our multiplex PCR approach works with two different polymerases with divergent sensitivity. In our experiments, Platinum SuperFi II DNA polymerase exhibited higher sensitivity than Q5 DNA polymerase; however, the CVR amplicon was not detected in most samples. This might be due to the lower melting temperature of the CVR primers, which might be incompatible with the annealing temperature (i.e., 60 °C) used. We therefore recommend designing new primers for the CVR target region to increase the melting temperature. Furthermore, our primer ratios were optimized for the Q5 DNA polymerase, so recalibration may increase the detection sensitivity of the CVR amplicon. Despite the higher detection sensitivity, it may still be necessary to increase the sequencing time, as the number of ASFV-specific reads was generally lower for the Platinum SuperFi II DNA polymerase compared to the Q5 DNA polymerase, especially in ASFV-positive samples with a low viral load. Increasing sequencing time may also help improve the consensus sequence of the intergenic region between *I73R* and *I329L* when using the Platinum SuperFi II DNA polymerase, as this was generally less accurate compared to sequences obtained with Q5 DNA polymerase. Thus, while our approach combined with Q5 DNA polymerase provides genomic information for all amplicons in samples with high to middle viral loads, Platinum SuperFi II DNA polymerase is better suited for samples with lower viral loads, although further optimization of our approach with this enzyme is required.

A higher detection limit for a broad range of ASFV variants is advantageous for the successful genomic characterization of ASFV, especially in the context of wild animals, where samples are primarily collected from carcasses. Unfortunately, due to national and international regulations in place for ASFV, it was not possible to evaluate the method on samples from naturally infected animals or other genotypes. Nevertheless, we demonstrate that our assay can, even with a limited number of samples, discriminate at least four genotypes (I, II, VIII, and X) and is applicable to samples from blood, liver, and lymph node biopsies, indicating its versatility across different specimen matrices. However, the lymph node samples used had low viral loads resulting in low or no detection of amplicons. Therefore, using other organs (i.e., spleen and lungs) as input for the multiplex PCR might be better suited to our approach, as these have been shown to have higher viral loads [40]. Whether this indeed is the case remains to be determined.

In conclusion, the multiplex approach described in this study effectively discriminates among the four ASFV genotypes tested and supports their subgrouping. While promising, the method requires further validation with a larger sample size and additional representative ASFV strains from the traditional 24 known ASFV genotypes to verify its full potential in enhancing ASFV strain identification and epidemiological tracing.

## Figures and Tables

**Figure 1 viruses-16-01460-f001:**
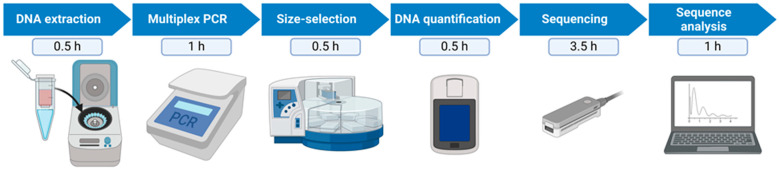
Schematic overview of the different workflow steps and approximate time needed to process 48 samples in parallel. Figure created with BioRender.com (https://app.biorender.com/, accessed on 13 May 2024).

**Figure 2 viruses-16-01460-f002:**
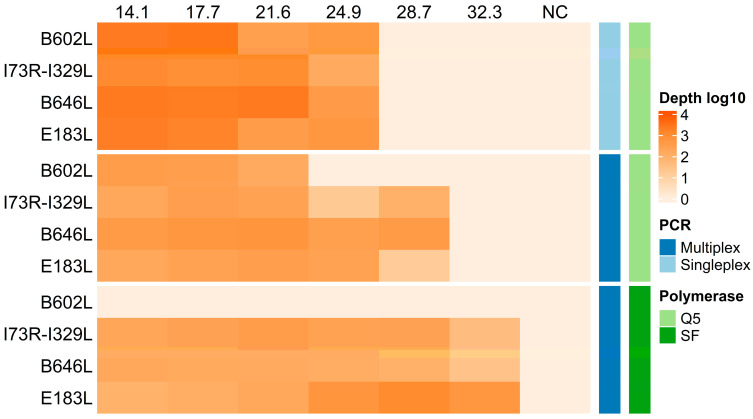
Sensitivity of singleplex and multiplex assays with Q5 DNA polymerase (Q5) and Platinum SuperFi II DNA polymerase (SF). Ten-fold serial dilutions of ASFV (with corresponding Ct values on the top of the graph, NC is the negative control) were tested using singleplex and multiplex PCR with the primer sets targeting four genomic regions (*B602L*, *I73R-I329L*, *B646L*, and *E183L*). The heat map shows the resulting mean sequencing depth in a log10 scale (gradient of orange, mean of three independent replicates). Corresponding mean and standard deviation are plotted in Appendix A.

**Figure 3 viruses-16-01460-f003:**
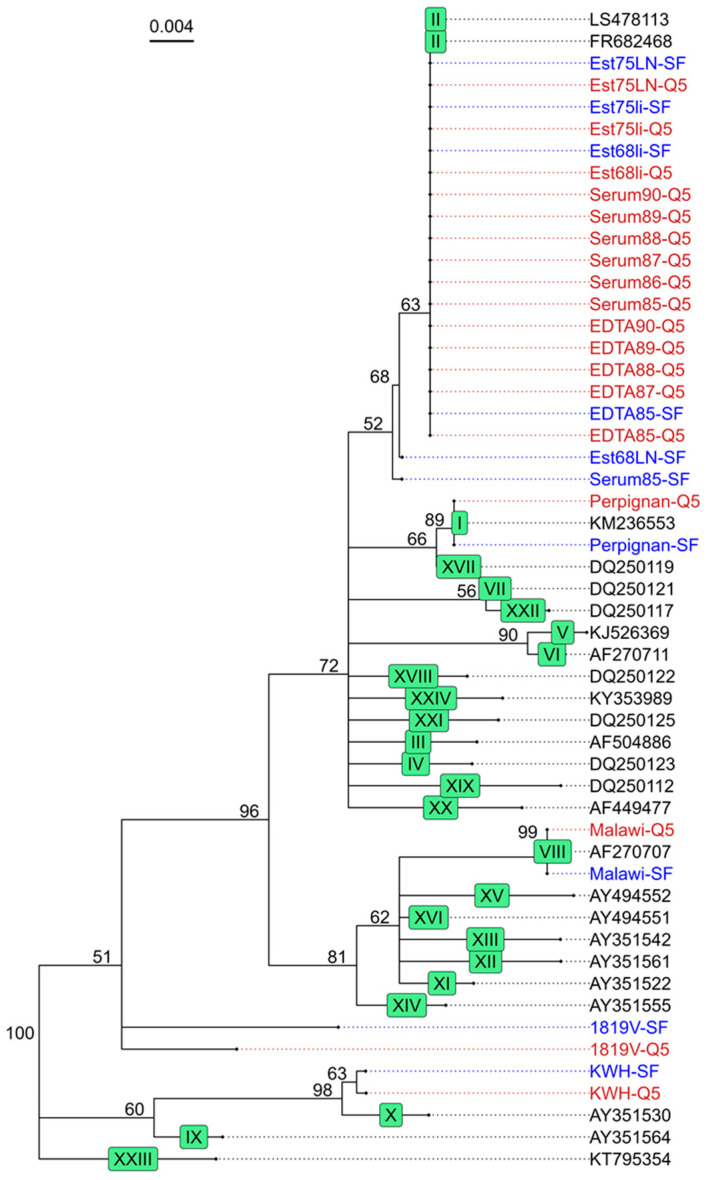
Phylogenetic tree based on the partial *B646L* gene sequence. To construct the phylogenetic tree, we aligned at least one representative sequence for each genotype (black, accession number as name) and the sequences generated in our study using the Q5 DNA polymerase (red) and the Platinum SuperFi II DNA polymerase (blue), using them as input for the Geneious Tree Builder. The samples sequenced in this study all clustered to the expected genotype (green box). All Georgia 2007 and Estonia 2014 samples cluster together with other genotype II ASFV strains (LS478113 and FR682468). The Perpignan strain clustered to KM236553, which is a genotype I ASFV strain, whereas Malawi and KWH clustered to AF270707 and AY351530, respectively, genotype VIII and X strains. Of note, the cell culture-adapted strain 1819V does not cluster with any known ASFV genotypes.

**Figure 4 viruses-16-01460-f004:**
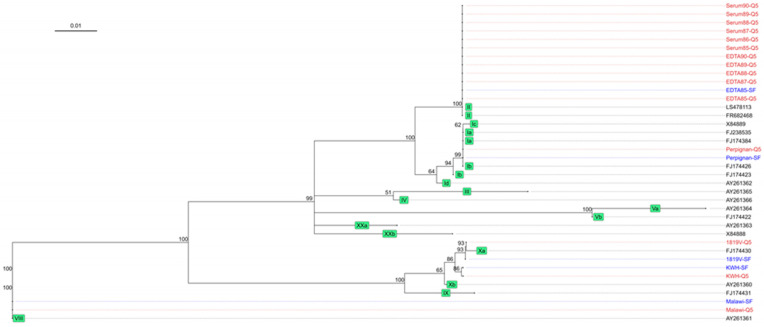
Phylogenetic tree based on the complete *E183L* gene sequence. The tree was constructed as described in Figure 3. The *E183L* amplicons corresponding to the genotype II ASFV samples sequenced in our study clustered to the other p54 genotype II strains (LS478113, FR682468). The ASFV genotype I strain Perpignan clustered closest to the p54 genotype I subgroup a (FJ238535, FJ174384). KWH clustered between genotype X subgroup a (FJ174430) and subgroup b (AY174431), whereas 1819V clustered closest to genotype X subgroup a. Finally, ASFV genotype VIII Malawi clustered closest to p54 genotype VIII AY261361.

**Figure 5 viruses-16-01460-f005:**
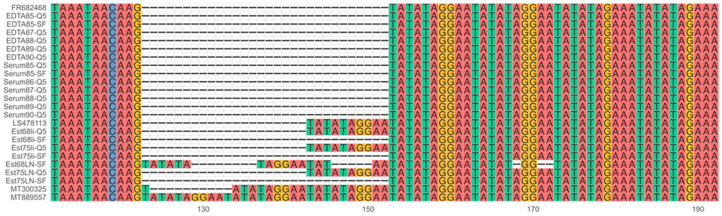
Multiple sequence alignment of the intergenic region between *I73R* and *I329L* genes (ECO1). To further discriminate genotype II ASFV, we aligned the *I73R-I329L* intergenic region sequences using the MAFFT aligner. The characteristic 10 nt long tandem-repeat sequence (TRS, TATATAGGAA) is repeated twice in the Georgia 2007 strain (IGR-I, FR682468), and the EDTA blood and serum samples show the same genotype. All the Estonia 2014 samples, amplified with the Q5 DNA polymerase, perfectly aligned to the Estonia 2014 reference (LS478113) in the IGR-II variant, which has the TRS repeated three times. The accession numbers MT300325 and MT889557 correspond to the variants IGR-III and IGR-IV with four and five times the TRS, respectively.

**Figure 6 viruses-16-01460-f006:**
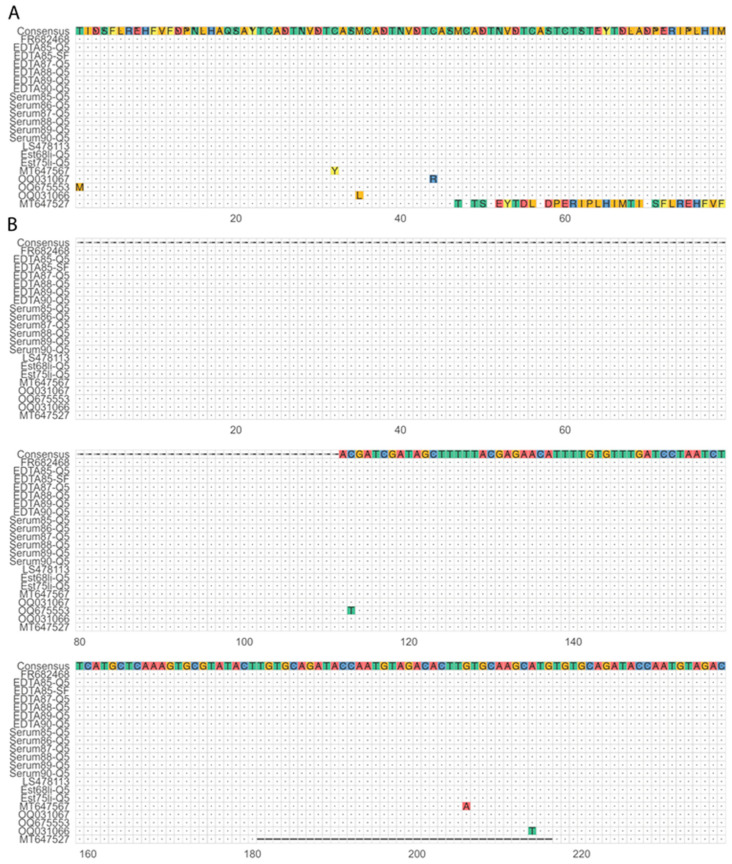
Multiple sequence alignments of the CVR region for genotype II sequences. (**A**) On the top of the alignment, the consensus sequence called from our samples and one representative for each known variant is represented. The translated CVR region has characteristic amino acid tetramer repeats, including some single nucleotide polymorphisms (colored amino acids), and in one variant (CVR2) a deletion of three amino acid tetramers (MT647527). All our samples belong to the variant CVR1, like the Georgia 2007 strain (FR682468). The other known CVR1 variants for genotype II ASFV are also aligned: CVR1-SNP1 (MT647567), CVR1-SNP2 (OQ031066), CVR1-SNP3 (OQ031067), and CVR1-SNP4 (OQ675553). (**B**) The nucleotide sequences of the same amplicon were also aligned, where all the samples sequenced in this study align without mismatches to the Georgia 2007 strain (FR682468).

**Table 1 viruses-16-01460-t001:** Overview table with the results of all samples for each DNA polymerase used. The serum, EDTA blood, lymph node (LN) and liver (li) samples are labeled with the respective animal number. The sequencing results are sorted according to their Ct value and divided by amplicon, where the samples are classified as follows: coverage depth higher than 10x (++), depth between 1 and 9 (+), no amplicon detected (-). Samples that were not amplified and sequenced are marked as not determined (ND).

	qPCR	Q5 DNA Polymerase				Platinum SuperFi II DNA Polymerase			
	Ct Value	*B646L*	*I73R-I329L*	*B602L*	*E183L*	*B646L*	*I73R-I329L*	*B602L*	*E183L*
Serum85	16	++	++	++	++	+	++	-	+
Serum90	16.1	++	++	++	++	ND	ND	ND	ND
Serum86	17.4	++	++	++	++	ND	ND	ND	ND
Serum87	17.4	++	++	++	++	ND	ND	ND	ND
Serum88	17.5	++	++	++	++	ND	ND	ND	ND
Serum89	18.2	++	++	++	++	ND	ND	ND	ND
KWH	19.1	++	++	++	++	++	++	++	++
EDTA87	19.4	++	++	++	++	ND	ND	ND	ND
EDTA90	19.4	++	++	++	++	ND	ND	ND	ND
EDTA88	19.5	++	++	++	++	ND	ND	ND	ND
EDTA89	19.6	++	++	++	++	ND	ND	ND	ND
EDTA85	19.7	++	++	++	++	++	++	++	++
1819V	20.2	++	++	++	++	++	++	+	++
Perpignan	21.3	++	++	++	++	++	++	-	++
Malawi	21.6	++	++	++	++	++	++	-	++
Est75li	25.7	++	++	++	-	++	++	+	+
Est68li	26.3	++	++	++	-	++	++	+	+
Est75LN	29.9	++	++	+	-	++	++	-	+
Est68LN	31.1	-	-	-	-	+	++	-	+

## Data Availability

The data presented in the study have been deposited in the European Nucleotide Archive (ENA) at EMBL-EBI under the accession number PRJEB75237 (https://www.ebi.ac.uk/ena/browser/view/PRJEB75237).

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
