# Peer review of "Multiplex PCR Approach for Rapid African Swine Fever Virus Genotyping"

_viruses, 2024, doi:10.3390/v16091460_

Round 1

Reviewer 1 Report (Previous Reviewer 1)

Comments and Suggestions for Authors

Dear authors,

The primary drawback of the manuscript remains the selection of the regions and the limited number of samples tested. To address this, the conclusion paragraph should be revised for clarity and accuracy. Currently, it states, "In conclusion, our multiplex approach can discriminate multiple genotypes and support subgrouping of ASFV strains." This should be updated to reflect the specific scope of the study. Here is a suggested version of the conclusion:

"In conclusion, the multiplex approach effectively discriminates among the four genotypes tested in this study and supports the subgrouping of ASFV strains. This targeted discrimination highlights the potential utility of the method for precise genotypic identification within the scope of the tested genotypes. While the findings are promising, the limited number of samples tested necessitates further validation with a broader sample set to confirm the robustness and generalizability of this approach. Future studies should aim to include a more extensive array of genotypes and a larger number of samples to enhance the reliability and applicability of this method for comprehensive ASFV strain differentiation and epidemiological investigations."

Author Response

Reviewer 1

We thank the reviewer for the thorough review of the article.

Dear authors,

The primary drawback of the manuscript remains the selection of the regions and the limited number of samples tested. To address this, the conclusion paragraph should be revised for clarity and accuracy. Currently, it states, "In conclusion, our multiplex approach can discriminate multiple genotypes and support subgrouping of ASFV strains." This should be updated to reflect the specific scope of the study. Here is a suggested version of the conclusion:

"In conclusion, the multiplex approach effectively discriminates among the four genotypes tested in this study and supports the subgrouping of ASFV strains. This targeted discrimination highlights the potential utility of the method for precise genotypic identification within the scope of the tested genotypes. While the findings are promising, the limited number of samples tested necessitates further validation with a broader sample set to confirm the robustness and generalizability of this approach. Future studies should aim to include a more extensive array of genotypes and a larger number of samples to enhance the reliability and applicability of this method for comprehensive ASFV strain differentiation and epidemiological investigations."

We thank the reviewer for the suggestion, and we adapted the concluding paragraph accordingly.

Reviewer 2 Report (Previous Reviewer 3)

Comments and Suggestions for Authors

This work describes an alternative applications to ASFv sequencing approach. The topic is very current and constantly evolving. Even if the gold standard is the Illumina sequence, alternative ways are always sought that can synergistically improve the reliability of the data. The work may be of some interest, although I have doubts about the use of these four molecular markers. it is also true that in the light of new knowledge the chosen markers may be revised. The work contains small ideas. This approach should probably be better investigated and applied to verify its effectiveness. At present, I do not consider it a first choice, but it is still useful to have a broad knowledge of all the molecular approaches that can be developed and which could provide an advantage in the sequencing of a virus as complex as that of African swine fever.

Author Response

Reviewer 2

We thank the reviewer for the thorough review of the article.

This work describes an alternative applications to ASFv sequencing approach. The topic is very current and constantly evolving. Even if the gold standard is the Illumina sequence, alternative ways are always sought that can synergistically improve the reliability of the data. The work may be of some interest, although I have doubts about the use of these four molecular markers. it is also true that in the light of new knowledge the chosen markers may be revised. The work contains small ideas. This approach should probably be better investigated and applied to verify its effectiveness. At present, I do not consider it a first choice, but it is still useful to have a broad knowledge of all the molecular approaches that can be developed and which could provide an advantage in the sequencing of a virus as complex as that of African swine fever.

We thank the reviewer for the insight, and positive feedback.

Reviewer 3 Report (Previous Reviewer 5)

Comments and Suggestions for Authors

Licheri et al, July 2024. Multiplex PCR approach for rapid African swine fever virus 2 genotyping.

The authors evaluate a set of 4 primers in multiplex format used for ASF genotyping, follow by nanopore ligation-based amplicon sequencing. The test is used to genotype ASF originated from different matrices and classified some genotypes and strains using a single PCR reaction. Optimization by swapping primers based on specific country / region making a versatile tool to speed up the processing time and lower costs during ASF outbreaks. 

For this resubmission, the authors improved the structure of the manuscript; by including a limit of detection / Sensitivity experiment, and the comparison with a second polymerase to increase Se for detection of amplicons when multiplexed. The data is of interest and can contribute to a more rapid identification and epidemiological tracing for ASFV. 

Please see some comments below to be addressed:

It will be of interest for the authors to include some comment on the introduction on the new classification of P72 based on whole gene sequence and if the method presented will be suitable for the new characterization.

Line 128: Please include on how the samples were loaded when multiplexed and single-plex. It will help if the authors can clarify in Methods Section, when the samples were single plex if they were loaded per individual dilution. As for multiplexing, please indicate if all libraries out of all PCR products for all 4 genes, corresponding to all dilutions were loaded together and if each ASF strain (Estonia or Georgia) was loaded together or in separate flow cell.

It will be great if the authors can expand if samples were loaded on the flow cells as multiplexed, meaning combining different pools of samples after libraries were prepared for all experiments presented on the manuscript.

Line 172: Please include the number of replicates done to determine the Sensitivity for each enzyme with each different strain.  What is the std deviation and how many replicates, to make the conclusions stronger (at least 3 replicates need to be included). 

Line 179: The authors mentioned a Sensitivity loss of 5 – 8-fold when multiplexing. Is this loss when multiplexing expected as per manufacturer protocol or due to combination of the primers and the thermocycling condition based on incompatible annealing temperature, as indicated on the Discussion for some of the genes targeted? (such as ECO1 and CVR)

Can the authors expand on why dilution -3 for B646L is negative and then dilution -4 shows 100 % coverage? Is this due to an error when pipetting, or due to lower Se when getting to higher dilutions? The addition of extra replicates, for a total of 3 repeats will be very helpful to evaluate repeatability and reproducibility as well as Se. Experimental design is good and helps determine Se of the test, as mentioned above, at least 3 replicates need to be included.

Line 180: will be good for the authors to expand more into the results when Platinum Polymerase is use targeting P72 and PPA, and add some %, to make the differences between both enzymes clearer.

Please include Ct values for the dilutions / samples corresponding to Figure 2.

 Can the authors explain why B602L gene is not showing any sequence? Is this due to lack of amplification? Is this case, is due to low reactivity of primer sets? Amplification of CVR seems to be very inconsistent through the samples, without much improvement when the Platinum Polymerase is evaluated, therefore adding some replicates can help to evaluate consistency and repeatability. 

Please can the authors expand why B602L is not amplified / sequenced for Estonia when multiplexing with Platinum enzyme.

Line 195: Sensibility seems to be higher when Platinum is used as shown by the authors in Table 1 for one sample (Est68LN). However, when looking at each gene individually, it seems to be an improvement for B602L detection when Q5 Pol is used as compared with Platinum. But the opposite when looking at E183L gene, improvement is seen with Platinum as compared with Q5. Will be great if the authors can expand a little more into these differences. 

Figure 3. Add genotypes in roman numbers to maintain consistency through the manuscript.

When characterizing sample 1819V, listed as “unknown”, the sample doesn’t cluster together with any of the know ASF genotypes, based on p72, however the authors can characterize when looking at other genes. Therefore, it may be of interest that the authors highlight that when working with unknown samples, the multiplex approach can help using different genes for further characterization when P72 may not perform.

On the previous submission, authors presented a set of 6 serum and blood samples, to evaluate that all 4 PCR products could be detected in both sample matrices with equal sequencing coverage and depth. It is not clear why those samples were removed from the manuscript, which are good references for repeatability, at least when samples are evaluated with Q5 polymerase.  

Line 219:  Please rewrite paragraph. You may want to consider dividing the paragraph in 2 sentences for a better understanding.

Platinum polymerase was included for comparison with Q5 Polymerase, with the aim to increase Se on the assay when multiplexing samples. It will be great if more detail on conclusions is included, since increase in Se doesn't seem to be consistent with the new Platinum enzyme and further optimization may still be needed. Some information on when one enzyme can be selected over the other one will be of interest. 

Author Response

Reviewer 3

We thank the reviewer for the thorough review of the article.

Licheri et al, July 2024. Multiplex PCR approach for rapid African swine fever virus 2 genotyping.

The authors evaluate a set of 4 primers in multiplex format used for ASF genotyping, follow by nanopore ligation-based amplicon sequencing. The test is used to genotype ASF originated from different matrices and classified some genotypes and strains using a single PCR reaction. Optimization by swapping primers based on specific country / region making a versatile tool to speed up the processing time and lower costs during ASF outbreaks.

For this resubmission, the authors improved the structure of the manuscript; by including a limit of detection / Sensitivity experiment, and the comparison with a second polymerase to increase Se for detection of amplicons when multiplexed. The data is of interest and can contribute to a more rapid identification and epidemiological tracing for ASFV.

Please see some comments below to be addressed:

It will be of interest for the authors to include some comment on the introduction on the new classification of P72 based on whole gene sequence and if the method presented will be suitable for the new characterization.

We thank the reviewer for the suggestion. However, because the focus of our study was on the evaluation of a PCR multiplex approach based on the primers from the European reference lab, and the relative recent publication of this approach, we decided to only expand on the possibility of using the new classification in combination with our method in the discussion.

Line 128: Please include on how the samples were loaded when multiplexed and single-plex. It will help if the authors can clarify in Methods Section, when the samples were single plex if they were loaded per individual dilution. As for multiplexing, please indicate if all libraries out of all PCR products for all 4 genes, corresponding to all dilutions were loaded together and if each ASF strain (Estonia or Georgia) was loaded together or in separate flow cell.

We thank the reviewer for the suggestion. We now added the information to the paragraph starting at line 134.

It will be great if the authors can expand if samples were loaded on the flow cells as multiplexed, meaning combining different pools of samples after libraries were prepared for all experiments presented on the manuscript.

We added this information to the paragraph starting at line 134.

Line 172: Please include the number of replicates done to determine the Sensitivity for each enzyme with each different strain.  What is the std deviation and how many replicates, to make the conclusions stronger (at least 3 replicates need to be included).

We repeated the sensitivity assay in triplicates using material from the Georgia strain for all PCR conditions and plotted the results in the new Figure 2 and described in the paragraph starting at line 182.

Line 179: The authors mentioned a Sensitivity loss of 5 – 8-fold when multiplexing. Is this loss when multiplexing expected as per manufacturer protocol or due to combination of the primers and the thermocycling condition based on incompatible annealing temperature, as indicated on the Discussion for some of the genes targeted? (such as ECO1 and CVR)

A multiplex PCR can lead to a decrease in sensitivity compared to a singleplex PCR for multiple reasons (i.e., competition for reagents, amplicon length, primer design and their efficiency at a specific annealing temperature (as indicated in the discussion for CVR)). In the revised manuscript we only see a loss in sensitivity for the CVR amplicon.

Can the authors expand on why dilution -3 for B646L is negative and then dilution -4 shows 100 % coverage? Is this due to an error when pipetting, or due to lower Se when getting to higher dilutions? The addition of extra replicates, for a total of 3 repeats will be very helpful to evaluate repeatability and reproducibility as well as Se. Experimental design is good and helps determine Se of the test, as mentioned above, at least 3 replicates need to be included.

This is most likely due to the lower sensitivity at higher dilutions. By using three replicates, we could confirm that this is not a consistent event.

Line 180: will be good for the authors to expand more into the results when Platinum Polymerase is use targeting P72 and PPA, and add some %, to make the differences between both enzymes clearer.

The paragraph was partially rewritten to reflect the new results from the sensitivity experiments and we mention both p72 and PPA amplicons.

Please include Ct values for the dilutions / samples corresponding to Figure 2.

The Ct values have been added to Figure 2.

 Can the authors explain why B602L gene is not showing any sequence? Is this due to lack of amplification? Is this case, is due to low reactivity of primer sets? Amplification of CVR seems to be very inconsistent through the samples, without much improvement when the Platinum Polymerase is evaluated, therefore adding some replicates can help to evaluate consistency and repeatability.

The absence of the B602L amplicons is most likely due to the lack of amplification. This is the case when using the Platinum SuperFi II DNA polymerase in combination with the CVR primer sets and most likely due to the annealing temperature (lines 365-370). We confirmed this during the revision.

Please can the authors expand why B602L is not amplified / sequenced for Estonia when multiplexing with Platinum enzyme.

We repeated the sensitivity assay in three replicates only for the Georgia samples for the updated manuscript. However, the most likely reason for failed amplification is the incompatibility between the Platinum SuperFi II DNA polymerase and the characteristics of the CVR primer set.

Line 195: Sensibility seems to be higher when Platinum is used as shown by the authors in Table 1 for one sample (Est68LN). However, when looking at each gene individually, it seems to be an improvement for B602L detection when Q5 Pol is used as compared with Platinum. But the opposite when looking at E183L gene, improvement is seen with Platinum as compared with Q5. Will be great if the authors can expand a little more into these differences.

These results reflect the results of the sensitivity assay, where the B602L is not amplified when using the Platinum SuperFi II DNA polymerase, whereas it can be amplified when using the Q5 DNA polymerase. As mentioned above, this is most likely due to the suboptimal primer annealing temperature. However, the buffer conditions of the Platinum SuperFI II DNA polymerase seem to improve the amplification efficiency of the other amplicons.

Figure 3. Add genotypes in roman numbers to maintain consistency through the manuscript.

The genotype numbering of Figure 3 and 4 has been changed.

When characterizing sample 1819V, listed as “unknown”, the sample doesn’t cluster together with any of the know ASF genotypes, based on p72, however the authors can characterize when looking at other genes. Therefore, it may be of interest that the authors highlight that when working with unknown samples, the multiplex approach can help using different genes for further characterization when P72 may not perform.

We thank the reviewer for the suggestion. We added it in lines 322-324.

On the previous submission, authors presented a set of 6 serum and blood samples, to evaluate that all 4 PCR products could be detected in both sample matrices with equal sequencing coverage and depth. It is not clear why those samples were removed from the manuscript, which are good references for repeatability, at least when samples are evaluated with Q5 polymerase. 

We added the 6 serum samples and the 5 EDTA samples for the Q5 DNA polymerase.

Line 219:  Please rewrite paragraph. You may want to consider dividing the paragraph in 2 sentences for a better understanding.

We rephrased the sentence to improve understanding.

Platinum polymerase was included for comparison with Q5 Polymerase, with the aim to increase Se on the assay when multiplexing samples. It will be great if more detail on conclusions is included, since increase in Se doesn't seem to be consistent with the new Platinum enzyme and further optimization may still be needed. Some information on when one enzyme can be selected over the other one will be of interest.

This information is present in the paragraph starting at line 364. We added some recommendation of when to use which enzyme at the end of the same paragraph as suggested by the reviewer.

Round 2

Reviewer 3 Report (Previous Reviewer 5)

Comments and Suggestions for Authors

Licheri et al, authors of the manuscript “Multiplex PCR approach for rapid African swine fever virus genotyping”, addressed all the comments presented by this reviewer. The final manuscript has been very much improved. The manuscript is well written, the results are well presented with detailed experimental design. This reviewer appreciates the consideration from the authors to address all the suggestions given. This reviewer recommends this manuscript to be considered for publication in Viruses.

Few comments:

Abstract Line 20. Add the 4 genes in parenthesis after the following sentence: "routinely used for ASFV genotyping (list genes in here)"...

M&M Line 80. Please include the list of organs used in parenthesis.

M&M Line 94. Replace “organ material” by “organ homogenate”, to keep consistency.

Figure 2. Please include in the legend that the values on top of the graph correspond to “Ct values”

Line 361. Add a comment such as “even with a limited number of samples” (since only one sample was run per genotype).

Author Response

Reviewer

We thank the reviewer for the thorough review of the article.

Licheri et al, authors of the manuscript “Multiplex PCR approach for rapid African swine fever virus genotyping”, addressed all the comments presented by this reviewer. The final manuscript has been very much improved. The manuscript is well written, the results are well presented with detailed experimental design. This reviewer appreciates the consideration from the authors to address all the suggestions given. This reviewer recommends this manuscript to be considered for publication in Viruses.

Few comments:

Abstract Line 20. Add the 4 genes in parenthesis after the following sentence: "routinely used for ASFV genotyping (list genes in here)"...

We added the genes in the abstract of the revised manuscript.

M&M Line 80. Please include the list of organs used in parenthesis.

This information was added to the revised manuscript.

M&M Line 94. Replace “organ material” by “organ homogenate”, to keep consistency.

We replaced the word as suggested by the reviewer.

Figure 2. Please include in the legend that the values on top of the graph correspond to “Ct values”

We updated the figure legend accordingly.

Line 361. Add a comment such as “even with a limited number of samples” (since only one sample was run per genotype).

We added the suggested sentence in the revised manuscript.

This manuscript is a resubmission of an earlier submission. The following is a list of the peer review reports and author responses from that submission.

Round 1

Reviewer 1 Report

Comments and Suggestions for Authors

The manuscript titled “Multiple PCR for rapid genotyping of African swine fever virus.” de Licheri et al., describes a multiplex PCR based on the four primer sets commonly used for African swine fever virus genotyping, followed by amplicon sequencing based on nanopore ligation. This method is intended to be used for genotyping. However, the work presented here, although as an idea it could be useful, is not well designed since it does not provide additional information to recent genotyping methods described in the literature.

There are major drawbacks that must be resolved before being considered for publication. The main one is linked to the regions used. Although these regions are useful for ASF genotyping, new regions have currently been described that allow the differentiation of up to 24 different subtypes within genotype 2 that have not been considered or are mentioned in the document. A more exhaustive review of the current situation of ASF in the world, in Europe and the genotyping methods currently described should have been done. The study is poorly planned from a conceptual point of view. From a methodology point of view, the selected samples do not include representative samples of all circulating genotypes, so it cannot be concluded that multiplex PCR could be used worldwide. Likewise, it does not amplify, in the case of the CVR region, when Ct values lower than 30 are obtained. The first thing that should have been done is to determine the sensitivity of the multiple technique compared to the individual PCR. But the main criticism is that it does not contribute anything new to what is currently published and that it allows outbreaks to be traced more effectively than the technique described. Other additional regions would have to be included.

Reviewer 2 Report

Comments and Suggestions for Authors

In this study, Licheri et al. evaluated a multiplex PCR assay with four primer sets from the European reference laboratory for ASFV genotyping, followed by nanopore ligation-based amplicon sequencing. And with this protocol they accurately classified multiple genotypes and strains of ASFV. This manuscript is generally well written and potentially valuable for ASF genotyping. However, the review has several concerns with the paper. 

1.   They should evaluate the sensitivity and specificity of the multiplex PCR.

2. Generally, the sensitivity of the multiplex PCR is much lower than the single PCR. So, actually I do not recommend multiple amplification for genotyping and sequencing. As shown in Table 1, it looks that the B602L gene could not be detected from samples with Ct value oved 20, it means the sensitivity is very low, and also E183L gene was undetectable in samples with Ct value over 26. Especially for weak positive samples, single PCR is preferred.

3. Most samples were from pigs experimentally infected with ASFV, or cell culture supernatants, they are also recommended to test natural infected pig samples.

Reviewer 3 Report

Comments and Suggestions for Authors

Reviewer 4 Report

Comments and Suggestions for Authors

I reviewed the manuscript entitled “Multiplex PCR for rapid African Swine Fever Virus genotyping” by Matthias Licheri, et al. In this manuscript authors evaluated a multiplex PCR based on the four primer-sets routinely used for ASFV genotyping, followed by Nanopore ligation-based amplicon sequencing. 

Overall, I think this it is an interesting concept/protocol, considering the great value of having a rapid, reliable diagnostic for this important disease. However, there are some issues that should be properly addressed by the authors.

 A)   The authors used only 21 samples to test the method. How do they determined that this number is the appropriated to validated it? In my opinion, the authors must include more samples.

B)   In the discussion, authors stated that this methodology works for different biological matrices (line 239). To conclude this, the authors must include more samples (as I mentioned in point A). 

C)   Regarding the statement in Line 245, about the multiplex approach can discriminate all genotypes, how the authors can assume this without including in their methodology samples from all the different genotypes? They only include genotype I, II, VIII and X. I am not agreeing with this. Although most of the current outbreaks has been related with genotype II, the methodology should include all the genotypes (as the authors are claiming). I recommend to the authors change the statement only referring specifically with the genotypes included in the study, or really include a broad number of samples to include more genotypes analysis.

D)   How do they authors validated their method? Since this is a partial genome analysis, the authors should include a validation with whole genome analysis, or another sequencing platform to validate their work.

Although this methodology can be very valuable, the authors need to improve the study. I highly recommend to the authors include more samples and a validation for their method (whole sequencing) to fulfill the requirements to publish the manuscript. That is why I am rejecting the manuscript in its present version.

Reviewer 5 Report

Comments and Suggestions for Authors

The manuscript by Licheri et al, describes a Multiplex PCR for rapid ASF genotyping, based on 4 primer sets used for genotyping, along with Nanopore sequencing allowing ASF identification within 5 hours. The authors described that this method could genotype ASF DNA from different biological matrices and classify multiple genotypes and strains using a single PCR reaction. Additionally, the authors suggest that the method can be optimized by adding or swapping primer sets based on needs, making it a rapid tool to reduce time and decrease cost of genotyping during an ASF outbreak.

It is well known the use of these primer sets recommended by the European reference laboratory for characterization of ASFV, as described by several authors. Therefore, the main novelty is this manuscript, is the implementation of nanopore sequencing along with these set of primers for a rapid turnaround time, making the manuscript more into a methods development or improvement of a protocol that is in use currently, such as Sanger or Illumina.

Comments respect to the Manuscript:

The manuscript needs to be improved to justify some of the conclusions the authors described, such as adding or swapping primer sets easily, matrices used and covering different genotypes, as well as total number of samples tested.

It will be good for the authors to expand on which are the options to swap primers and pool reactions and how easily this can be in terms of evaluating different concentrations and thermocycling conditions. In the event of swapping primers, optimization of the conditions will need to be described to justify this as an option for this method, since different pools of primers may not always have a successful output.

For evaluation of the different products, it will be of interest to include each gene individually amplified and run on a gel. This will allow to discriminate better each PCR product, and to have a clear image of different products with no overlapping. Sequencing of each product individually will be useful and will add to this information for confirmation.

What is the sensitivity of the technique compared with real time PCR or any other sequencing methods used, such as Sanger, in combination with these amplicons, for each gene targeted? What is the viral titer or Ct value, using p72, recommended for this technique to be successful? Adding information on the LOD, using reference viral stocks with known titer and Ct values will be of great interest and will add into the manuscript.

Line 140.  Is the B602L band corresponding to 780 bp no visible on the gel since is not present or due to low Sensitivity of the PCR?

Please improve the legend on Figure 2 to described what is shown in here.

Described into detail and add references related with difference on size of the same genes throughout the different samples, is this expected? Please include references for product size expected if this has been reported in the literature.

Include MW for ladder in Figure 2.

Paragraph lines 142 – 147, add reference to Figure 2.

The authors indicated that the multiplex PCR has been evaluated only on GII ASF samples, this corresponds to a pig experimentally infected with Georgia 2007. It will be of interest to evaluate individual primers with different genotypes as done with GII, to assure they are working properly.

Will be of interest to evaluate the cell culture viruses spiked into different matrices, such as blood, serum, or homogenates of naïve animals, to evaluated how the test performs for different genotypes when spiked in different matrices, mimicking a real sample.

Seems no bands are detected for ECO1 and CVR when KWH and Malawi were used. Can the authors expand on possible reasons for these results?

Differences in CVR product when comparing Serum and EDTA-blood samples are shown on the gels. Is the difference due to different inputs used for PCR or expression of the different genes?

LINE 147. Please include information of coverage and depth per region amplified.

Line 148. Authors discuss the challenge to obtain wild boar carcass samples. This sentence doesn’t seem to be related in here. Please introduce into context to understand why this sample type can be of interest and role of wild boar in ASF. Authors can add this information in the introduction.

Line 157. Presence of virus in different tissues varies and this is reflected with a difference in Ct values, additionally Se of the test may be low and not able to detect above a Ct of 26 – 30. Therefore, non-detection may be related with the Se of the test to detect the virus, will be good for this to be evaluated and described in the manuscript under LOD studies and Sensitivity along with sample type, as mentioned above.

Table 1. B602L doesn’t seem to have a good Se when run on gel. qPCR needs to be included for all different genes, for better Se and accuracy on the test used as reference and to determine limit of detection. Even though, how the authors explain lack of this gene on conventional PCR, in samples with Ct values of 21, such as Malawi and Perpignan?

Line 160. Table 1: Please include in Methods qPCR used and presented in Table 1.

The experiments do not include enough samples to justify the authors conclusions and the number of genotypes tested is limited. Matrices used are EDTA blood and serum from ASFV Georgia experimentally inoculated pigs and limited number of tissues from animals experimentally infected with ASFV Estonia, both viruses corresponding to Genotype II. It is not clear why the authors conclude that this method allows subgrouping of especially GI and II, when only one cell culture ASF GI sample was evaluated on the manuscript.

When characterizing sample 1819V, listed as “unknown”, the sample doesn’t cluster together with any of the know ASF genotypes, based on p72. This information is not in agreement with previous data, where a Ct value of 20.2 for p72 and a high coverage (20X) was obtained for this sample. However, when E182L gene is sequenced, this sample seems to cluster with Genotype X. Please can you expand on why this isolate is not clustered with any known ASF genotypes when tested with p72?

What is coverage, depth and cutoff to consider data confident to make a call on a variant or SNP when using this method?

In this manuscript the main novelty is the use of the nanopore as a rapid method for turnaround time. Therefore, it will be of interest for the authors to expand into the discussion on the advantages of this platform, which is not mentioned throughout the manuscript. Some information such as of pros /cons, multiplex capabilities as compared with other platforms, error rates, depth and coverage can add more into the manuscript. It is mentioned that PCR / nanopore is cheaper and more affordable than Sanger / NGS, however no numbers are presented. It will be of interest to include some information to compare price per sample.

Authors need to be more precise when mentioning that this method can discriminate all genotypes when only 4 out of the 24 genotypes were evaluated in h manuscript and with very limited number of samples.

Adding a timeline comparison of this methodology vs Sanger or Illumina will be of interest. As well as a comparison of the sequences obtained with this methods vs nanopore. 

Generation of amplicons by PCR is cheaper and faster, but these data is not providing the user with the information that is available from the whole genome. It will be important for this to be included in the discussion.